# Exploring the Associated Factors of Depression, Anxiety, and Stress among Healthcare Shift Workers during the COVID-19 Pandemic

**DOI:** 10.3390/ijerph19159420

**Published:** 2022-08-01

**Authors:** Norsham Juliana, Nor Amira Syahira Mohd Azmi, Nadia Effendy, Nur Islami Mohd Fahmi Teng, Sahar Azmani, Nizam Baharom, Aza Sherin Mohamad Yusuff, Izuddin Fahmy Abu

**Affiliations:** 1Faculty of Medicine and Health Sciences, Universiti Sains Islam Malaysia, Nilai 71800, Malaysia; amirasyahira188@gmail.com (N.A.S.M.A.); nadia@usim.edu.my (N.E.); drazmanisahar@usim.edu.my (S.A.); drnizamb@usim.edu.my (N.B.); azasherin@usim.edu.my (A.S.M.Y.); 2Faculty of Health Sciences, Universiti Teknologi MARA, Puncak Alam 42300, Malaysia; nurislami@uitm.edu.my; 3Institute of Medical Science Technology, Universiti Kuala Lumpur, Kajang 43000, Malaysia; izuddin@unikl.edu.my

**Keywords:** shift workers, psychosocial, physical activity, eating habits, COVID-19

## Abstract

Background: The recent pandemic of COVID-19 has had a tremendous impact on healthcare frontliners. This study sought to assess healthcare shift workers’ depression, anxiety, and stress and its associated factors. Methods: The sampling frame includes healthcare shift workers directly managing COVID-19 cases around Klang Valley, Malaysia. The participants’ mental health status was assessed using the Depression, Anxiety, Stress Scale-21 (DASS-21). The associated factors specified in this study include sleep quality, physical activities, and eating habits. Pearson’s χ^2^ and simple and multivariable binary logistic regression models were constructed following the Hosmer–Lemeshow approach to determine the potential associated factors. Results: A total of 413 participants were recruited. Overall, 40.7% of participants had one or more symptoms of depression, anxiety, or stress. Poor sleep quality was significantly associated with all mental health outcomes of depression, anxiety, and stress. Inactivity was found to be strongly associated with symptoms of depression and anxiety. At the same time, eating habits were strongly associated with anxiety and stress. Conclusions: Sleep quality, inactivity, and eating habits that were found to be associated with the mental health status of healthcare shift workers are modifiable factors that must be addressed to curb mental health issues among this group of workers.

## 1. Introduction

Shift work is often associated with jobs that require work outside the traditional daytime hours of 8 a.m. to 5 p.m. It has become pervasive across economically developed countries such as Canada, the United Kingdom, France, Russia, and Malaysia, leading to many arising issues such as mental health problems, extreme fatigue, poor health, disorganised households, and abandoned children or partners. By working nights or early morning shifts, a person must be awake when the circadian drive for alertness is low and asleep when it is high, which is opposite to the natural biological rhythm [1]. This will affect a person’s circadian rhythm and disrupt major systems controlled by this rhythm, such as metabolic health, heart health, cancer risks, and mental health. Disruption of the circadian rhythm does not only contribute to many diseases but also may eventually result in increased errors in the workplace and poor work outcomes [2].

The COVID-19 pandemic has overwhelmed most countries’ public health sectors and capacities. The first case detected in Malaysia was on 24 January 2020, and the Restriction of Movement Order was implemented on 18 March 2020. COVID-19 has caused a significant increase in work demand among healthcare workers, which consequently caused them to pause in perplexity at managing their mental health. During the increasing trend of confirmed positive cases, healthcare shift workers continue to serve our country in various roles of surveillance, screening, diagnosis, and treatment. However, they also face many issues, such as a shortage of personal protective equipment (PPE), hospital beds, and other medical equipment while responding to the pandemic healthcare needs. This indirectly places them in severe burnout experience, which leads to stress, depression, and anxiety [3]. A study by the Ministry of Health’s Institute for Health Behavioral Research reported that about 14.2% of healthcare workers suffered severe mental disorders during the COVID-19 pandemic. They also reported that 24.7% lacked childcare support at home while 16.7% lacked moral support at work [4].

Several studies conducted during the pandemic showed alarming levels of psychological issues among the community that could be associated with losing family members to COVID-19, movement restrictions, jobs, and financial problems [5,6]. Following these issues, it was enlightening to study the psychological effects of the pandemic among healthcare shift workers, as they are the frontliners who had to deal with the wreaked chaos of the pandemic. Studies on working hours among healthcare workers during the pandemic have shown that those working longer hours had alarming burnout and stress scores. In addition, longer hours are associated with prolonged contact with patients or samples, prolonged time of wearing PPE, sleep deprivation, and poor eating habits [3,7,8]. These psychological effects among healthcare workers can be exacerbated by multiple risk factors such as gender, medical condition, financial status, and occupational differences.

A study conducted in 2018 reported that approximately 38% of healthcare workers in Kuala Lumpur and Selangor were found to have moderate to severe depression. Bivariate analysis of this data showed that males, assistant medical officers, and working more than 10 h per day were the significant risk factors [9]. Amid the pandemic, anxiety, depression, and stress among healthcare shift workers increased, leading to many suicidal cases. Healthcare shift workers have to face tremendous workloads and long working hours while battling the fear of being infected with COVID-19. This is in line with other studies that reported that healthcare workers, especially those working in shifts, are more likely to be hospitalised for COVID-19 [10]. In addition, recent data reported that sleep disruptions are associated with the likelihood of testing positive for COVID-19 and its complications [11]. Furthermore, they have a constant fear of infecting colleagues and family members, which consequently burdens their mental health. Recent studies reported that occupational differences are one of the risk factors in which nurses and laboratory technicians handling COVID-19 samples had higher levels of psychological distress than other healthcare workers. Meanwhile, another study by Ilhan and Kupeli (2022) found that healthcare workers with financial difficulties were at the highest risk for developing anxiety, depression, and stress [12].

Studies by Teo et al., (2022) and Subhas et al., (2021) focused on healthcare shift workers’ psychological well-being and its associated factors [13,14]. However, to the best of our knowledge, there are no studies in Malaysia to date that include sleep quality, physical activity and eating habits as part of the factors to be associated with the psychological well-being of healthcare shift workers during the COVID-19 pandemic. Therefore, this study aimed to determine the psychological status of healthcare shift workers and the related factors in our country.

## 2. Materials and Methods

### 2.1. Study Design and Subjects

This cross-sectional study involved healthcare workers aged 19 to 60 years old. The sampling frame includes healthcare workers who directly managed COVID-19 cases in hospitals around Klang Valley and worked three shifts for at least one year. The sample of this study included the participants from hospitals around Klang Valley because in Malaysia, Klang Valley reported the highest cases of COVID-19. There are fourteen listed government hospitals in Klang Valley, and we chose the hospitals that directly managed COVID-19 cases during the study period.

Participants were randomly recruited using stratified random sampling. Inclusion criteria included literacy in Malay or English language and healthcare workers with Malaysian citizenship. At the same time, those diagnosed with sleeping disorders and/or mental illness were excluded from this study.

Sample size calculation was performed using the OpenEpi toolkit for proportion, with a 95% confidence interval (CI), significance level (*p*) of 0.05, and two sides, with a degree of accuracy of 5%. A previous study by Teixeira et al., (2020) highlighted the prevalence ratio of physical activity with psychosocial problems among shift workers. Hence, 392 participants were required for this study [15]. 

### 2.2. Ethical Aspects of the Study

This study obtained the approval of the Medical Research and Ethics Committee and National Medical Research Registry with code reference NMRR-19-2796-50756. Informed consent was obtained from all subjects prior to the study.

### 2.3. Questionnaires and Data Collection

The components of socio-demographic data, shift work schedule information, and self-reported anthropometry measurement were included in the first part of the questionnaires. The subsequent part of the questionnaires focused on the components of physical activity, eating habits, and psychosocial well-being that comprised mental health, quality of life, sleep quality and work engagement. Four validated questionnaires adopted include the International Physical Activity Questionnaire-Short form (IPAQ) [16], Dutch Eating Behaviour Questionnaire (DEBQ) [17], Pittsburgh Sleep Quality Index (PSQI) [18], and Depression, Anxiety, and Stress Scale 21 (DASS-21) [19]. The authors of this study included subject matter experts of the study. Therefore, the questionnaires were arranged and selected based on expert consensus. The average duration of completing the questionnaires was ten to fifteen minutes.

Data collection was conducted from February 2020 to August 2021 during the pandemic of COVID-19. All questionnaires were self-administered. Those who fulfilled the inclusion criteria were provided with a set of questionnaires in the Malay version, either in printed material or via an online platform. The anthropometric measurement consisted of self-reported height and body weight. Based on the self-reported data, the participant’s body mass index (BMI) was calculated and classified into different categories following the World Health Organisation (WHO) classification. 

### 2.4. Statistical Analysis

Results were analysed using the IBM Statistical Package for Social Science (SPSS) version 26.0 (SPSS Inc., Chicago, IL, USA). Pearson’s χ^2^ and simple and multivariable binary logistic regression models were constructed following the Hosmer–Lemeshow approach to determine the potential associated factors. Associations between factors and outcomes were presented as odds ratio (OR) with 95% confidence intervals. A *p* value of <0.05 was considered statistically significant.

The response rate to the questionnaires was 95.6%. Correspondingly, the number of participants meets the sample size requirement. We managed to collect 430 participants. However, we only included 413 participants in the analysis. Of 413 participants, 123 answered the printed questionnaires and 290 responded to the online questionnaires.

## 3. Results

### 3.1. Background of the Participants

A total of 413 participants participated in this study. The mean age of the participants was 31.7 ± 5.9 years, and most participants were less than 40 years old at the time of data collection (89.1%). The majority of the participants were nurses (60.8%), women (81.1%), and received tertiary education (90.5%). Table 1 refers to the background of the participants and further details.

### 3.2. Mental Health Status

Table 2 describes the findings from the mental health status of the participants. Overall, 40.7% of participants had one or more symptoms of mental health problems of either depression, anxiety, or stress at different levels (mild, moderate, severe, or extremely severe). The majority of participants had moderate depression (15.5%), moderate anxiety (20.6%), and mild stress (5.6%), with mean Depression, Anxiety, Stress Scale-21 (DASS-21) scores of 15.61 ± 5.7, 13.89 ± 5.5, and 21.04 ± 5.3, respectively. Furthermore, Figure 1 portrays the mental health status of the participants in this study.

### 3.3. Sleep Quality, Physical Activities and Eating Habits of the Participants

In terms of the participants’ sleep quality, most of them had poor sleep quality (58.1%) with a mean Pittsburgh Sleep Quality Index (PSQI) score of 6.49 ± 3.1. Regarding physical activity, IPAQ data revealed that most participants were either inactive (31.7%) or minimally active (43.6%). The general question of physical activity on intentional exercise is also in accord with the International Physical Activity Questionnaire—Short Form (IPAQ) data, with more than half (61.3%) of participants not conducting any intentional exercise. Finally, the eating habits of participants were classified based on the Dutch Eating Behaviour Questionnaire (DEBQ). The majority of the participants obtained low scores for emotional eating (86.9%), with a cut-off point of 3.25. Conversely, participants scored high external eating (78.0%) and restraint eating (63.4%) with a cut-off point of 2.5.

### 3.4. Factors Associated with Depression, Anxiety and Stress of the Participants

Factors associated with the participants’ mental health are highlighted in Table 3, Table 4 and Table 5, specifically addressing depression, anxiety, and stress. The results are presented in crude odds ratio (OR) and adjusted odds ratio (AOR).

Factors strongly associated with symptoms of depression include BMI categorized as obese (AOR = 2.18; 95% CI: 1.1–4.2), inactivity (AOR = 2.16; 95% CI: 1.1–4.1) and poor sleep quality (AOR = 2.33; 95% CI: 1.4–3.9). Other factors such as age group of fewer than 40 years old (OR = 2.54; 95% CI: 1.0–6.2), marital status of being single (OR = 1.77; 95% CI: 1.1–2.8), and healthcare position as nurses (OR = 0.50; 95% CI: 0.3–0.8) were also found to be associated with the symptom of depression when analysed independently.

In terms of anxiety, the participants who were less than 40 years old had triple the chances of having anxiety symptoms (AOR = 3.29; 95% CI: 1.3–8.5). There were twice the odds of having anxiety symptoms for those who were inactive (AOR = 2.00; 95% CI: 1.1–3.7). Those with poor sleep quality had the odds of two times greater anxiety symptoms (AOR = 2.09; 95% CI: 1.3–3.3). Furthermore, independent analyses showed that women had increased odds of having anxiety symptoms (OR = 1.83; 95% CI: 1.0–3.2). A similar trend was found in participants categorized with emotional and external eating habits (OR = 2.09; 95% CI: 1.2–3.7); (OR = 1.77; 95% CI: 1.0–3.0).

Strong predictors of stress include poor sleep quality (AOR = 3.96; 95% CI: 1.7–9.1) and low scores of restrained eating habits (AOR = 2.94; 95% CI: 1.42–5.0). Other independent factors include marital status of single (OR = 2.73; 95% CI: 1.5–5.0) and healthcare position as nurses (OR = 0.47; 95% CI: 0.2–0.9).

## 4. Discussion

The demographic data showed that the over-representation of females in this study (81.1%) is consistent with previous studies among shift workers and healthcare workers [20,21]. In accord with other studies, this study focused predominantly on a specific occupational group of healthcare workers. This eliminates any bias on the nature of the shift schedule that may vary depending on the nature of the workplace [22,23]. The age group of participants was also mainly concentrated among those younger than 40 years old (89.1%). Beyond the age of 40, healthcare workers are upgraded to a higher position that adapts office hours scheduling with calls. The department that extensively adopted the three-shift working hours was the emergency department [24]. Therefore, most of the participants came from this department. Overall, the prevalence of overweight and obesity among participants was high, with 43.8% categorised in this group. Local studies conducted regionally also reflected a similar trend, with a reported range of 29–49% prevalence of overweight and obese among healthcare workers [25,26].

The emergence of the COVID-19 pandemic has had an unprecedented impact on the health systems and the mental health and well-being of the frontliners. There was reported evidence of adverse mental health issues brought by the pandemic to healthcare workers, suggesting that dealing with the impact will be a continuous post-pandemic process [27,28,29]. Shanafelt et al., (2019) reported that studies between 2011 and 2017 in the United States found that burnout is already a pre-existing problem among healthcare workers either working regular hours or shifts [30]. During the pandemic that has caused more than 5.5 million deaths worldwide to date (28 June 2022), challenges for healthcare workers were mounting as they were expected to be effective in giving service. Even though the participants in this study were recruited during the pandemic, they were involved directly as frontliners and were working in hospitals with a high density of patient inflow; the prevalence of depression and anxiety is lower compared to other global and local studies [31,32,33,34]. Stress, however, showed a higher prevalence reported in this study compared to a recent local study by Salaton et al., (2022) but was on par with findings in a systematic review and meta-analysis conducted by Li et al., (2021) [33,34].

Working in the healthcare industry itself is challenging. Thus, working in shifts in the industry will add another level of overwhelming pressure and challenge for an individual. For those with standard sleep patterns, daytime wakefulness is driven by the biological clock, which produces circadian rhythmicity, driving increased alertness during the daytime and decreased attention during the night [1,35]. Shift workers involved in night shifts are reported to be more susceptible to psychological and mental health issues than those working regular hours. The problems include irritability, somatisation, obsessive-compulsive disorder, interpersonal sensitivity, anxiety, altered mood, and paranoid disorders, which were predominantly prevalent [1]. All these underlying issues may explain the high prevalence of stress found in this study, as the participants are healthcare workers working three shifts.

Findings from this study support previous studies that indicate the average sleep quality among night shift workers to be relatively lower compared with day and non-night rotating workers [36,37]. Overall, participants’ mental health status is associated with their sleep quality, with those with poor sleep quality having higher odds of disruptive mental health status. Working night shift is proven to tamper with cortisol and melatonin rhythms, the hormones strongly associated with sleep quality [38]. A common problem that shift workers share is a lack of adequate sleep following their night shifts. An epidemiological study revealed that 90% of individuals with symptoms of depression have sleep disturbances [39]. Similarly, a study by Oh et al., (2019) reported a strong association between sleep quality and symptoms of depression and anxiety. The strong association between sleep quality and mental health warrants serious attention for future intervention [40,41]. This effort will further curb the deteriorating effect of mental health issues faced by healthcare shift workers.

Physical activity has long been associated with good psychological outcomes, with its neurophysiological effects influencing important neural mechanisms related to depressive and anxiety disorders [42]. The pandemic has resulted in the absence or reduction of physical activity, further elevating the risk of mental issues such as anxiety and depression. Furthermore, having an increased workload among healthcare workers during the pandemic also contributes to the possible factor of reducing time for physical activity beyond working hours [43]. This association between physical activity and depression and anxiety symptoms has a possible bidirectional relationship. Either the physical activity reduces the risk of anxiety and/or depression symptoms, or the symptoms may also lead to reduced physical activities [44].

Previous studies that associate symptoms of depression with obesity pointed out that usually the association occurs at a young age or in young adulthood [45,46,47]. Coherent findings encountered in this study reflected the pool of participants aged less than 40 years old. A previous study by Luppino et al., (2010) showed that obesity contributed to a 55% increased risk of developing depression symptoms over time [48]. Over the last decade, meta-analyses and research findings reported that obesity and depression are linked with specific gender of women, and this relationship was inversely proportional in men [49,50,51]. Therefore, findings in this study support a similar theory, as most of the participants are women. Possible explanations for the findings of this study include the drastic change in work demand and lifestyle factors that may play a prominent role in the association observed. Restriction of outdoor activities may lead to an increase in body mass index (BMI). Thus, it inevitably predisposes individuals to symptoms of depression due to a reduction in activity, social interaction, and diversity of daily life [45].

Mohd Azmi et al., (2020) described that the duration of time breaks among shift workers significantly affects their eating habits [1]. Time restriction impedes the regular timing of a vital eating schedule for well-being. A restrained eater is defined as an individual with intense dieting or restrictive food intake of specific macronutrients or food types with sporadic overeating episodes. This eating habit is not a protective mechanism towards ideal body weight. However, it was found to be associated with an increased risk of overweight and obesity [52,53]. Similar to the explanation of other factors, the relationship of this factor of restrained eating habits with stress symptoms can be bi-directional. The stressful environment during a pandemic may result in irregular eating patterns connected to restrained eating or vice versa [52].

The strength of this study was that we could gain all data homogenously during the pandemic stage of COVID-19. Therefore, the data reflect the actual psychological state and its factors in relation to the time of disaster. Moreover, the selected centres were hospitals with a high density of daily patients. The significant predictors gained from this study are important parameters to be highlighted to policymakers in order to develop preventive intervention strategies to preserve shift worker well-being. Future development of occupational health modules must include recommendations on gaining good sleep quality, physical activity and eating habits among shift workers.

Part of the limitations of this study was the restriction of sample collection via a face-to-face method. Face-to-face interviews allow two-way communication between the researcher and the subject matter. Online platforms restrict communication, but in order to overcome this limitation, we highlighted the contact number and email addresses of the researchers on the online platform so that the participants may contact them if facing any problems. In addition, this study only examined the healthcare shift workers in Malaysia, which may not represent the condition of the workers in other regions. The nature of shift schedules among healthcare workers may vary based on policy and local regulations of each country. Therefore, results from this study must be carefully adapted, as they may not reflect healthcare shift workers with different types of shift schedules. Based on Moyo et al., (2022), the frequency of contracting COVID-19 among healthcare workers influenced their psychological status [54]. However, the parameters were not part of the questionnaires in the current study. Hence, we suggest that future studies also include this parameter.

Additional future studies with multiple types of shift workers in various industries should be included in order to determine whether other industries share similar predictive factors in maintaining psychological well-being.

## 5. Conclusions

In summary, this study proves the significant factors of sleep quality, physical activity and eating habits to be associated with the mental health of healthcare shift workers. It shows that undesirable symptoms of depression, anxiety, and stress among healthcare shift workers are associated with poor sleep quality, probably aggravated by poor biological synchronisation of the sleep and wake cycle. Inactivity is also an essential factor that is strongly associated with symptoms of depression and anxiety. The pandemic may enhance participants’ perceived depression and anxiety, as the high density of patients require them to be dynamically moving at work. Furthermore, factors of obesity and eating habits associated with symptoms of depression or stress, respectively, are equally important to be further elucidated in the future. The results of this study contribute towards specific modifiable factors. In addition, the novelty of this study is that it is the first study in Malaysia to date to include these predictors as part of the factors associated with the psychological well-being of healthcare shift workers during the COVID-19 pandemic. It is useful to be a guidance for future research to develop interventions such as lifestyle module recommendations, particularly on sleep quality, physical activity and eating habits for the psychological maintenance of healthcare shift workers. The direction of future policy on occupational health also may focus on these factors in maintaining the psychological well-being of shift workers. Further investigation must focus on a longitudinal study to confirm the causal relations between these factors and healthcare shift workers’ mental health status.

## Figures and Tables

**Figure 1 ijerph-19-09420-f001:**
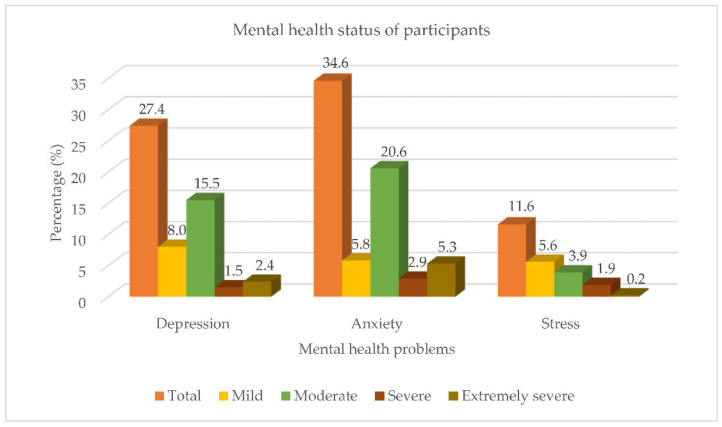
Mental health status of participants. Note: Participants may have more than one symptom of mental health problems.

**Table 1 ijerph-19-09420-t001:** Background of participants.

	*n*	%
Age		
<40 years old	368	89.1
≥40 years old	43	10.4
Gender		
Men	78	18.9
Women	335	81.1
Ethnicity		
Malays	337	81.6
Chinese	12	2.9
Indians	47	11.4
Others	17	4.1
Educational status		
Secondary education	27	6.5
Post-secondary education	12	2.9
Diploma	231	55.9
Bachelor/postgraduate	143	34.6
Marital status		
Single	135	32.7
Married	275	66.6
Divorced/separated/widowed	3	0.7
Household income *^1^		
Low (<MYR 4850)	192	46.5
Middle (MYR 4850–RM 10,959)	207	50.1
High (≥MYR 10,960)	14	3.4
Healthcare position		
House officer	47	11.4
Medical officer	91	22
Nurse	251	60.8
Paramedics	24	5.8
Department		
Emergency and trauma	175	42.4
Medical-based	125	30.3
Surgical-based	113	27.4
Part-time job involvement *^2^		
No	369	89.3
Yes	44	10.7
Comorbidities		
No	353	85.5
Yes	60	14.5
Smoking/vaping status		
No	390	94.4
Yes	23	5.6
Alcohol consumption		
No	399	96.6
Yes	14	3.4
Body mass index (BMI) *^3^		
Underweight	30	7.3
Normal	202	48.9
Overweight	107	25.9
Obese	74	17.9

* Note: ^1^ Based on the Department of Statistics Malaysia official portal classification. ^2^ Part-time job involvement; includes participants involved in other part-time work after their working hours. ^3^ Body mass index (BMI) of the participants based on the WHO classification; underweight (<18.5 kg/m^2^); normal (18.5–24.9 kg/m^2^); overweight (25.0–29.9 kg/m^2^), obese (≥30.0 kg/m^2^).

**Table 2 ijerph-19-09420-t002:** Mental health status of participants.

Mental Health Status	*n* (%)	Mean ± S.D.
DASS-21		
Depression		
Mild (10–13)	33 (8.0)	15.61 ± 5.7
Moderate (14–20)	64 (15.5)	
Severe (21–27)	6 (1.5)	
Extremely severe (28+)	10 (2.4)	
Anxiety		
Mild (8–9)	24 (5.8)	13.89 ± 5.5
Moderate (10–14)	85 (20.6)	
Severe (15–19)	12 (2.9)	
Extremely severe (20+)	22 (5.3)	
Stress		
Mild (15–18)	23 (5.6)	21.04 ± 5.3
Moderate (19–25)	16 (3.9)	
Severe (26–33)	8 (1.9)	
Extremely severe (34+)	1 (0.2)	

**Table 3 ijerph-19-09420-t003:** The crude odds ratio (OR) and adjusted OR (AOR) for DASS-21 of Depression.

Factors (*n*)	DASS-21 Depression					
	Normal (*n*)	Depression (*n*)	Crude OR (95% CI)	*p* Value	Adjusted OR (95% CI)	*p* Value
Age group						
<40 years old	260 (70.8%)	107 (29.2%)	2.54 (1.0–6.2)	*p* = 0.041 *	1.63 (0.6–4.3)	*p* = 0.319
≥40 years old	37 (86.0%)	6 (14.0%)	Ref		Ref	
Gender						
Male	57 (73.1%)	21 (26.9%)	Ref		Ref	
Female	242 (72.5%)	92 (27.5%)	1.03 (0.6–1.8)	*p* = 0.912	1.39 (0.7–2.9)	*p* = 0.374
Marital status						
Married	209 (76.3%)	65 (23.7%)	Ref		Ref	
Single	87 (64.4%)	48 (35.6%)	1.77 (1.1–2.8)	*p* = 0.012 *	1.41 (0.8–2.4)	*p* = 0.207
Divorced/separated/widowed	3 (100.0%)	0 (0.0%)	0.00 (0.0–0.0)	*p* = 0.999	0.00 (0.0–0.0)	*p* = 0.999
Healthcare position						
Medical officer	57 (62.6%)	34 (37.4%)	Ref		Ref	
House officer	29 (61.7%)	18 (38.3%)	1.04 (0.5–2.1)	*p* = 0.914	1.16 (0.5–2.6)	*p* = 0.711
Nurse	193 (77.2%)	57 (22.8%)	0.50 (0.3–0.8)	*p* = 0.008 *	0.63 (0.3–1.2)	*p* = 0.162
Paramedics	20 (83.3%)	4 (16.7%)	0.34 (0.1–1.1)	*p* = 0.064	0.40 (0.1–1.5)	*p* = 0.162
Body mass index (BMI)						
Underweight	20 (66.7%)	10 (33.3%)	1.48 (0.7–3.4)	*p* = 0.350	1.51 (0.6–3.8)	*p* = 0.380
Overweight	80 (75.5%)	26 (24.5%)	0.96 (0.6–1.7)	*p* = 0.890	1.24 (0.7–2.3)	*p* = 0.479
Obese	48 (64.9%)	26 (35.1%)	1.60 (0.9–2.8)	*p* = 0.106	2.18 (1.1–4.2)	*p* = 0.018 *
Normal	151 (74.8%)	51 (25.2%)	Ref		Ref	
Category of physical activity (IPAQ)						
Inactive	81 (61.8%)	50 (38.2%)	2.35 (1.3–4.3)	*p* = 0.005 *	2.16 (1.1–4.1)	*p* = 0.019 *
Minimally active	137 (76.5%)	42 (23.5%)	1.17 (0.6–2.1)	*p* = 0.607	1.16 (0.6–2.2)	*p* = 0.652
HEPA active	80 (79.2%)	21 (20.8%)	Ref		Ref	
Emotional eating habit (DEBQ)						
Low score	265 (74.0%)	93 (26.0%)	Ref		Ref	
High score	34 (63.0%)	20 (37.0%)	1.68 (0.9–3.1)	*p* = 0.092	1.54 (0.8–3.0)	*p* = 0.207
External eating habit (DEBQ)						
Low score	72 (79.1%)	19 (20.9%)	Ref		Ref	
High score	227 (70.7%)	94 (29.3%)	1.57 (0.9–2.7)	*p* = 0.115	1.62 (0.8–3.1)	*p* = 0.146
Restraint eating habit (DEBQ)						
Low score	106 (70.2%)	45 (29.8%)	Ref		Ref	
High score	193 (73.9%)	68 (26.1%)	0.83 (0.5–1.3)	*p* = 0.412	0.76 (0.4–1.3)	*p* = 0.292
Sleep quality (PSQI)						
Good	141 (82.5%)	30 (17.5%)	Ref		Ref	
Poor	157 (65.4%)	83 (34.6%)	2.49 (1.5–4.0)	*p* < 0.001 *	2.33 (1.4–3.9)	*p* = 0.001 *

* significant to *p* value < 0.05. Adjusted OR, results were adjusted for demographic factors.

**Table 4 ijerph-19-09420-t004:** The crude odds ratio (OR) and adjusted OR (AOR) for DASS-21 of Anxiety.

Factors (*n*)	DASS-21 Anxiety					
	Normal (*n*)	Anxiety (*n*)	Crude OR (95% CI)	*p* Value	Adjusted OR (95% CI)	*p* Value
Age group						
<40 years old	230 (62.7%)	137 (37.3%)	3.67 (1.5–8.9)	*p* = 0.004 *	3.29 (1.3–8.5)	*p* = 0.014 *
≥40 years old	37 (86.0%)	6 (14.0%)	Ref		Ref	
Gender						
Male	59 (75.6%)	19 (24.4%)	Ref		Ref	
Female	210 (62.9%)	124 (37.1%)	1.83 (1.0–3.2)	*p* = 0.035 *	1.65 (0.8–3.4)	*p* = 0.177
Marital status						
Married	186 (67.9%)	88 (32.1%)	Ref		Ref	
Single	80 (59.3%)	55 (40.7%)	1.45 (0.9–2.2)	*p* = 0.086	1.24 (0.7–2.1)	*p* = 0.414
Divorced/separated/widowed	3 (100.0%)	0 (0.0%)	0.00 (0.0–0.0)	*p* = 0.999	0.00 (0.0–0.0)	*p* = 0.999
Healthcare position						
Medical officer	61 (67.0%)	30 (33.0%)	Ref		Ref	
House officer	26 (55.3%)	21 (44.7%)	1.64 (0.8–3.4)	*p* = 0.178	1.53 (0.7–3.4)	*p* = 0.287
Nurse	162 (64.8%)	88 (35.2%)	1.11 (0.7–1.8)	*p* = 0.701	1.37 (0.7–2.6)	*p* = 0.338
Paramedics	20 (83.3%)	4 (16.7%)	0.41 (0.1–1.3)	*p* = 0.128	0.59 (0.2–2.1)	*p* = 0.423
Body mass index (BMI)						
Underweight	16 (53.3%)	14 (46.7%)	1.51 (0.7–3.3)	*p* = 0.293	1.67 (0.7–4.0)	*p* = 0.251
Overweight	81 (76.4%)	25 (23.6%)	0.53 (0.3–0.9)	*p* = 0.021 *	0.62 (0.4–1.1)	*p* = 0.107
Obese	44 (59.5%)	30 (40.5%)	1.18 (0.7–2.0)	*p* = 0.553	1.52 (0.8–2.8)	*p* = 0.175
Normal	128 (63.4%)	74 (36.6%)	Ref		Ref	
Category of physical activity (IPAQ)						
Inactive	79 (60.3%)	52 (39.7%)	1.80 (1.0–3.2)	*p* = 0.040 *	2.00 (1.1–3.7)	*p* = 0.029 *
Minimally active	115 (64.2%)	64 (35.8%)	1.53 (0.9–2.6)	*p* = 0.123	1.60 (0.9–2.9)	*p* = 0.112
HEPA active	74 (73.3%)	27 (26.7%)	Ref		Ref	
Emotional eating habit (DEBQ)						
Low score	242 (67.6%)	116 (32.4%)	Ref		Ref	
High score	27 (50.0%)	27 (50.0%)	2.09 (1.2–3.7)	*p* = 0.013 *	1.78 (0.9–3.4)	*p* = 0.074
External eating habit (DEBQ)						
Low score	68 (74.7%)	23 (25.3%)	Ref		Ref	
High score	201 (62.6%)	120 (37.4%)	1.77 (1.0–3.0)	*p* = 0.034 *	1.66 (0.9–3.0)	*p* = 0.096
Restraint eating habit (DEBQ)						
Low score	97 (64.2%)	54 (35.8%)	Ref		Ref	
High score	172 (65.9%)	89 (34.1%)	0.93 (0.6–1.4)	*p* = 0.733	0.84 (0.5–1.4)	*p* = 0.486
Sleep quality (PSQI)						
Good	127 (74.3%)	44 (25.7%)	Ref		Ref	
Poor	141 (58.8%)	99 (41.3%)	2.03 (1.3–3.1)	*p* = 0.001 *	2.09 (1.3–3.3)	*p* = 0.002 *

* significant to *p* value < 0.05. Adjusted OR, results were adjusted for demographic factors.

**Table 5 ijerph-19-09420-t005:** The crude odds ratio (OR) and adjusted OR (AOR) for DASS-21 of Stress.

Factors (*n*)	DASS-21 Stress					
	Normal (*n*)	Stress (*n*)	Crude OR (95% CI)	*p* Value	Adjusted OR (95% CI)	*p* Value
Age group						
<40 years old	321 (87.5%)	46 (12.5%)	2.94 (0.7–12.6)	*p* = 0.146	1.21 (0.3–5.7)	*p* = 0.814
≥40 years old	41 (95.3%)	2 (4.7%)	Ref		Ref	
Gender						
Male	67 (85.9%)	11 (14.1%)	Ref		Ref	
Female	297 (88.9%)	37 (11.1%)	0.76 (0.4–1.6)	*p* = 0.455	1.11 (0.4–2.8)	*p* = 0.828
Marital status						
Married	252 (92.0%)	22 (8.0%)	Ref		Ref	
Single	109 (80.7%)	26 (19.3%)	2.73 (1.5–5.0)	*p* = 0.001 *	2.05 (1.0–4.2)	*p* = 0.050
Divorced/separated/widowed	3 (100.0%)	0 (0.0%)	0.00 (0.0–0.0)	*p* = 0.999	0.00 (0.0–0.0)	*p* = 0.999
Healthcare position						
Medical officer	76 (83.5%)	15 (16.5%)	Ref		Ref	
House officer	37 (78.7%)	10 (21.3%)	1.37 (0.6–3.3)	*p* = 0.489	1.72 (0.6–4.7)	*p* = 0.287
Nurse	229 (91.6%)	21 (8.4%)	0.47 (0.2–0.9)	*p* = 0.035 *	0.74 (0.3–1.8)	*p* = 0.500
Paramedics	22 (91.7%)	2 (8.3%)	0.46 (0.1–2.2)	*p* = 0.327	0.36 (0.1–2.1)	*p* = 0.255
Body mass index (BMI)						
Underweight	25 (83.3%)	5 (16.7%)	1.64 (0.6–4.7)	*p* = 0.361	1.56 (0.5–5.0)	*p* = 0.456
Overweight	95 (89.6%)	11 (10.4%)	0.95 (0.4–2.0)	*p* = 0.890	1.47 (0.6–3.4)	*p* = 0.377
Obese	64 (86.5%)	10 (13.5%)	1.28 (0.6–2.8)	*p* = 0.547	2.11 (0.8–5.3)	*p* = 0.110
Normal	180 (89.1%)	22 (10.9%)	Ref		Ref	
Category of physical activity (IPAQ)						
Inactive	108 (82.4%)	23 (17.6%)	1.74 (0.8–3.8)	*p* = 0.158	1.44 (0.6–3.4)	*p* = 0.405
Minimally active	165 (92.2%)	14 (7.8%)	0.69 (0.3–1.6)	*p* = 0.389	0.61 (0.2–1.5)	*p* = 0.282
HEPA active	90 (89.1%)	11 (10.9%)	Ref		Ref	
Emotional eating habit (DEBQ)						
Low score	318 (88.8%)	40 (11.2%)	Ref		Ref	
High score	46 (85.2%)	8 (14.8%)	1.38 (0.6–3.1)	*p* = 0.439	1.35 (0.5–3.4)	*p* = 0.522
External eating habit (DEBQ)						
Low score	82 (90.1%)	9 (9.9%)	Ref		Ref	
High score	282 (87.9%)	39 (12.1%)	1.26 (0.6–2.7)	*p* = 0.554	1.76 (0.7–4.3)	*p* = 0.215
Restraint eating habit (DEBQ)						
Low score	124 (82.1%)	27 (17.9%)	Ref		Ref	
High score	240 (92.0%)	21 (8.0%)	0.40 (0.2–0.7)	*p* = 0.003 *	0.34 (0.2–0.7)	*p* = 0.003 *
Sleep quality (PSQI)						
Good	163 (95.3%)	8 (4.7%)	Ref		Ref	
Poor	200 (83.3%)	40 (16.7%)	4.08 (1.9–9.0)	*p* < 0.001 *	3.96 (1.7–9.1)	*p* = 0.001 *

* significant to *p* value < 0.05 Adjusted OR, results were adjusted for demographic factors.

## Data Availability

Not applicable.

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
