# Peer review of "Exploring the Associated Factors of Depression, Anxiety, and Stress among Healthcare Shift Workers during the COVID-19 Pandemic"

_ijerph, 2022, doi:10.3390/ijerph19159420_

Round 1
Reviewer 1 Report
This study attempts to assess healthcare shift workers' depression, anxiety, and stress and its associated factors. The subject is very interesting and actual. The paper is very well-written and discussed. However, the unique main problem is the language. Also, for the presentation of the results, it would be interesting to have some charts. Also, the discussion must present the advantages and disadvantages of this study in relation with others.
Reviewer 2 Report
1. On lines 85-91, “no studies in Malaysia up to date are focusing on the healthcare shift workers’ psychological well-being and its associated factors during the COVID-19 pandemic…” was stated; however, several studies found from the database also reported the psychological wellbeing of the healthcare workers with shift working schedule during COVID-19 in Malaysia, for example, the works of Teo et al. (2022) and Subhas et al. (2021). I do believe that there are some differences between yours and theirs. Thus, you have to point out the uniqueness and contributions of your study in lieu of stating “no studies…”.
Teo I, Nadarajan GD, Ng S, Bhaskar A, Sung SC, Cheung YB, Pan FT, Haedar A, Gaerlan FJ, Ong SF, Riyapan S, Do SN, Luong CQ, Rao V, Soh LM, Tan HK, Ong MEH. The Psychological Well-Being of Southeast Asian Frontline Healthcare Workers during COVID-19: A Multi-Country Study. International Journal of Environmental Research and Public Health. 2022; 19(11):6380. https://doi.org/10.3390/ijerph19116380
Subhas N, Pang NT-P, Chua W-C, Kamu A, Ho C-M, David IS, Goh WW-L, Gunasegaran YI, Tan K-A. The Cross-Sectional Relations of COVID-19 Fear and Stress to Psychological Distress among Frontline Healthcare Workers in Selangor, Malaysia. International Journal of Environmental Research and Public Health. 2021; 18(19):10182. https://doi.org/10.3390/ijerph181910182
2. In Table 1, why did “age” only have 2 subgroups? Why the subgroups were not “18-29, >29-39, …”? Based on what rationales that you made this classification for the age group? Any references for such categorisation? Why not set the cut-off of age as 50-year-old as 50-year-old and above workers are considered as older workers (De Lange et al., 2006, Fleming et al., 2007, Jones et al., 2013, Zwerling et al., 1996)?
3. As referred to Table 1, the sample size included both part-time and full-time workers. However, the sample size of part-time workers is a lot smaller than the full-time workers. It is better to exclude the part-time workers so as to eliminate the bias induced by the data of part-time workers.
4. It is believed that this study must have a number of limitations. Nevertheless, none of them were mentioned in the discussion. For example, the sample size was not large enough to reflect all target populations even if the sample size calculation was provided. This study only examined the night-shift healthcare workers in Malaysia which may not represent the condition of the workers in other regions. Thus, the data should be used with caution. Also, the working cultures are different among different regions which may reflect the actual situation of the workers in other regions too.
5. Any practical implications? This study provides a myriad of information on the psychological health of shift healthcare workers. It is believed that some suggestions on alleviating the mental health of the workers can be provided.
6. Regarding the study design of the age range included in this study, why those aged 18 were excluded?
Reviewer 3 Report
Dear Authors,
First of all, you need to define paper objectives (in the Introduction section). You don't need to speculate by stating at this point "The findings of this study may help to ensure the safety of healthcare shift workers". In the same time, in the Introduction, I was expecting to find the description of the essential parts of the paper with some clarifications.
Secondly, I suggest to insert a Literature review section where you could include some of the sources from the Introduction section but also some researches mentioned in the Materials and Methods section. In order to help you more, I listed here few titles that could offer some info related to your work:
1.Elbay, RY; Kurtulmuş, A.; Arpacıoğlu, S.; Karadere, E. Depression, anxiety, stress levels of physicians and associated factors in Covid-19 pandemics. Psychiatry Res. 2020 Aug; 290:113-130. doi: 10.1016/j.psychres.2020.113130
2.Arafa, A.; Mohammed, Z.; Mahmoud, O.; Elshazley, M.; Ewis A. Depressed, anxious, and stressed: What have healthcare workers on the frontlines in Egypt and Saudi Arabia experienced during the COVID-19 pandemic? J Affect Disord. 2021 Jan 1; 278:365-371. doi: 10.1016/j.jad.2020.09.080.
3. Magnavita, N.; Tripepi, G.; Di Prinzio, R.R. Symptoms in Health Care Workers during the COVID-19 Epidemic. A Cross-Sectional Survey. Int. J. Environ. Res. Public Health 2020, 17, 5218. https://doi.org/10.3390/ijerph17145218
4.Kapetanos, K.; Mazeri, S.; Constantinou, D.; Vavlitou, A.; Karaiskakis, M.; Kourouzidou, D. et al. (2021) Exploring the factors associated with the mental health of frontline healthcare workers during the COVID-19 pandemic in Cyprus. PLoS ONE 16(10): e0258475. https://doi.org/10.1371/journal.pone.0258475
5. Alnazly, E.; Khraisat, OM.; Al-Bashaireh, AM,; Bryant CL. (2021) Anxiety, depression, stress, fear and social support during COVID-19 pandemic among Jordanian healthcare workers. PLoS ONE 16(3): e0247679. https://doi.org/10.1371/journal.pone.0247679
6. Matarazzo, T.; Bravi, F.; Valpiani, G.; Morotti, C.; Martino, F.; Bombardi, S.; Bozzolan, M.; Longhitano, E.; Bardasi, P.; Roberto, D.V.; Carradori, T. CORONAcrisis—An Observational Study on the Experience of Healthcare Professionals in a University Hospital during a Pandemic Emergency. Int. J. Environ. Res. Public Health 2021, 18, 4250. https://doi.org/10.3390/ijerph18084250
Why these papers could be important? Because are related with current study and it represents experiences from other countries and could confirm or maybe refutes certain findings of the current paper.
Now, even though you delivered info related to the study designs, subjects and questionnaires I ask you to add few more things. For instance, the sample of your survey includes people from hospitals around Klang Valley. Why this region? (is a relevant area in terms of medical issues and COVID-19 treated cases?), How many hospitals are in that area?, All the hospitals are treated especially COVID-19 cases?, What was the initial number of participants (intended group)?, What was the response rate to the questionnaires?
Related to the questionnaires is important to know: Who designed the final form?, How many questions are included?, It was a pretest survey (in order to refine the questions)? Which was the survey period? Which was the average duration of completing the questionnaire responses? How many questionnaires were completed physically and how many online?
Reaching the end, some clarifications are needed. For example, in the Discussion section it is necessary to include comparative studies from different countries (maybe regions, countys, big cities) in order to confirm or refute the results obtained. On the other hand, it is necessary to emphasize the main limitations of the study.
The least well represented part is the Conclusions. Here it is necessary to further explain and develop the findings, to mention What is actually the original contribution made by the authors to the field (which brought concretely and as a novelty, the analysis made by the authors)?, How the results of this particular study can be considered useful for potential interested readers? etc.
Reviewer 4 Report
The work presented is of enormous interest and relevance. It is worth paying attention to the symptoms of depression, anxiety and stress experienced by healthcare workers in order to seek strategies to mitigate them.
However, there are a number of aspects that the authors should review in the submitted manuscript:
- In the abstract, authors should note the country in which the city in which they are conducting the research work is located. Also that the sample consisted mainly of nurses.
- Shift work varies from country to country. In the paper they point to the 8 to 5 pm schedule but in Spain, for example, the daytime schedule is 8 to 3 pm. This deserves to be explained in the introduction.
- Ethical aspects should be included in a specific section.
- The work of line 105 should be correctly referenced.
- It would be advisable for the authors to simplify tables 3, 4 and 5 to make them easier to read.
- It would be necessary to include at the end of the discussion a section on the limitations of the study and another on possible future lines of research.
Reviewer 5 Report
Dear Authors,
It is such a great honor to read your paper, please find some comments kindly below.
- In the introduction part, you did a great job on background of pandemic and health issue of shift workers.
- In the questionnaires and data collection, you mentioned 7 questionnaires, but only list 5 questionaries, it is necessary to double check the number of questionnaires you used in the research. Besides, inclusion discharge standard did not show clearly in your paper, we recommend you show illustrate the number of participants and the collection number you used in the analyze.
- This is no doubt that Dutch Eating Behavior Questionnaire (DEBQ) show the result of shift workers’ eating habit, however, could you please consider that is this questionnaire benefit for workers in the certain region?
- Regarding the table you list in this paper, we highly recommend that make the table more clearly and easier to read, it is better to delete some criteria which not necessary to your research result.
- In the discussion part, please show more findings from your research, and highlight more literatures which other scholars had been proved before, trying to find the gap of your findings.
We keen to hear other thoughts from your research, if you have any question, please let us know.
Kind Regards,
Yibo
Round 2
Reviewer 1 Report
The paper is improved, and it can be accepted.
Reviewer 3 Report
Dear Authors,
The answers given are mostly satisfactory. The points treated more superficially are 5 (partially) and 6. What I commented initially (to these points) was even more than you added. For instance in the last section (Conclusions) you actually added a phrase that does not even remotely answer the requests from the previous review. The questions were very clearly formulated: What is actually the original contribution made by the authors to the field? Which brought concretely and as a novelty, the analysis made by the authors? How can the results of this particular study be considered useful for potentially interested readers?
The value and importance of the paper were lost the most through the Conclusions section (the weak point of the article, the weakest area represented by ideas and important things).
Reviewer 5 Report
Thanks for inviting me again to evaluate the revised version of manuscript ijerph-1830957 entitled "Exploring The Associated Factors For Depression, Anxiety, And Stress Among Healthcare Shift Workers During Covid-19 Pandemic". The revised paper is well-written and is acceptable for being published.
